# Gács-Körner Common Information Variational Autoencoder

**Michael Kleinman**[1]  **Alessandro Achille**[2,3*]  **Stefano Soatto**[1]  **Jonathan C. Kao**[1]
[1]University of California, Los Angeles  [2]AWS AI Labs  [3]Caltech
michael.kleinman@ucla.edu    aachille@amazon.com
soatto@cs.ucla.edu    kao@seas.ucla.edu

## Abstract

We propose a notion of common information that allows one to quantify and separate the information that is shared between two random variables from the information that is unique to each. Our notion of common information is defined by an optimization problem over a family of functions and recovers the Gács-Körner common information as a special case. Importantly, our notion can be approximated empirically using samples from the underlying data distribution. We then provide a method to partition and quantify the common and unique information using a simple modification of a traditional variational auto-encoder. Empirically, we demonstrate that our formulation allows us to learn semantically meaningful common and unique factors of variation even on high-dimensional data such as images and videos. Moreover, on datasets where ground-truth latent factors are known, we show that we can accurately quantify the common information between the random variables.[2]

## 1 Introduction

Data coming from different sensors often capture information related to common latent factors. For example, many animals have two eyes that capture different but highly-correlated views of the same objects in the scene. Similarly, sensors of different modalities, such as eyes and ears, capture correlated information about the underlying scene, as do videos and other time series, where the sensors are separated in time rather than in modality. Learning how information of one sensor maps to information of another provides a self-supervised signal to disentangle the variability that is intrinsic in a sensor from the latent causes (e.g., objects) that are shared between multiple sensors. Indeed, there is evidence that infants spend a long time during development purposefully experiencing objects through different senses at the same time [1].

Motivated by this, we propose to learn meaningful representations of multi-view data by quantifying and exploiting such structure in a self-supervised fashion by using an information theoretic notion of *common information* as the guiding signal to disentangle common shared information present in high dimensional sensory data (Fig. 1).

However, defining a notion of common information is itself not trivial. The most natural and typical way to quantify the "common part" between random variables would be by quantifying their mutual information. But mutual information has no clear interpretation in terms of a decomposition of random variables in unique and common components. In particular, [2] note that there is generally no way to write two variables $X$ and $Y$ using a three part code $(A, B, C)$ such that $X = f(A, C)$, $Y = g(B, C)$ and where $C$ encodes all and only the mutual information $I(X; Y)$. Discovering the

---

*Work performed as external collaboration not related to Amazon

[2]Code available at: https://github.com/mjkleinman/common-vae

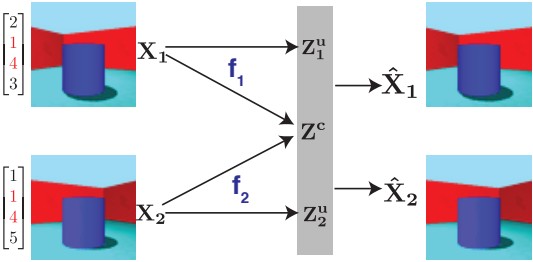

Figure 1: **High level schematic of approach.** Red denotes shared latent factors (size, shape, floor, background and object color) and black denotes unique latent (viewpoint). The aim is to extract $z_c$, which is a random variable that is a function of both inputs $x_i$. We also allow for unique latent variables $z_u$ to capture information that unique to each view. The latent representations are used to reconstruct the inputs.

largest common factor $C$, which encodes what is known as the Gács-Körner common information, from high dimensional data is then a distinct problem on its own [3, 4].

To the best of our knowledge, there are currently no approaches to compute or approximate the Gács-Körner common information from high-dimensional samples [4]. In this work, we seek to learn common representations that satisfy the constraint that they are (approximately) a function of each input. A contribution in this paper is that we generalize the constraint that the representation needs to be a deterministic function and allow it to be a stochastic map. As we later show, this is helpful for quantifying and interpreting the latent representation, and allows us to parameterize the optimization with deep neural networks.

We show that our objective can be optimized using a multi-view Variational Auto-Encoder (VAE). Since in general each view can contain individual factors of variation that are not shared between the views, we augment our model with a set of unique latent variables that can capture unexplained latent factors of variation, and show that the common and unique component can be efficiently inferred from data through standard training. While training the multi-view VAE, we simultaneously develop a scalable approximation for the Gács-Körner common information, as we describe in Section 3.

To empirically evaluate the ability to separate the common and unique latent factors we introduce two new datasets, which extend commonly used datasets for evaluating disentangled representations learning: dSprites [5] and 3dShapes [6]. For each dataset, we generate a set of paired views $(x_1, x_2)$ such that they share a set of common factors. This allows us to quantitatively evaluate the ability to separate the common and unique factors. We also compare our method to multi-view contrastive learning [7] and show that thanks to our definition we avoid learning degenerate representations when the views share little information.

We hypothesize that a key reason precluding the identification of latent generating factors from observed data is that receiving a single sample of a scene is quite limiting. Indeed, classical neuroscience experiments has shown that the ability to interact with an environment, as opposed to passively observing sensory inputs, is critical for learning meaningful representations of the environment [8].

## 2   Preliminaries and Related Work

The entropy $H(x)$ of a random variable $x$ is $\mathbb{E}_{p(x)}[\log \frac{1}{p(x)}]$. The mutual information $I(x; z) = H(z) - H(z|x)$. Another useful identity for mutual information that we use is $I(x; z) = \mathbb{E}_{p(x)}[KL(p(z|x)||p(z))]$ where KL denotes the Kullback-Leibler divergence.

**Gács-Körner Common Information.**   The Gács-Körner common information [2] is defined as

$$C_{GK}(x_1; x_2) := \max_z H(z) \quad \text{s.t.} \quad z = f(x_1) = g(x_2), \tag{1}$$

where $f$ and $g$ are *deterministic* functions. The Gács-Körner common information is thus defined through a random variable $z$ that is a deterministic function of both inputs $x_1$ and $x_2$. Among all

such random variables, $z$ is the random variable with maximum entropy. This has also been referred to as the "zero error information" in applications to cryptography [3]. It is an attempt to formalize and operationalize the idea of the common part between sources, which mutual information lacks. It is also a lower bound to the mutual information [2, 3]. To the best of our knowledge, there are no efficient techniques for computing the GK common information for high-dimensional $x_1, x_2$. We elaborate on the difference between GK common information and mutual information in App. E.

**Variational Autoencoders**   Variational Autoencdoers (VAEs) [9] are latent variable generative models that are trained to maximize the likelihood of the data by maximizing the *evidence lower bound*, or minimizing the loss:

$$\mathcal{L}_{\text{VAE}} = \mathbb{E}_{p(x)}[\, \mathbb{E}_{q_\phi(z|x)}[-\log p_\theta(x|z)] + KL(q_\phi(z|x) \,||\, p(z))]. \tag{2}$$

The VAE loss can be motivated in an information theoretic manner as optimizing an Information Bottleneck [10], where the reconstruction term encourages a *sufficient* representation and the KL regularization term encourages a *minimal* representation [11, 12]. The addition of a parameter $\beta$ to modify how the KL regularization is penalized leads to the following loss (and corresponds to the traditional VAE loss when $\beta = 1$):

$$\mathcal{L}_{\beta\text{-VAE}} = \mathbb{E}_{p(x)}[\, \mathbb{E}_{q_\phi(z|x)}[-\log p_\theta(x|z)] + \beta KL(q_\phi(z|x) \,||\, p(z))]. \tag{3}$$

For larger values of $\beta$ the representations become more disentangled, shown analytically in [13] and empirically in [14], although reconstructions become worse.

**Disentangled representations**   A guiding assumption for representation learning is that the observed data $x$ (i.e an image) can be generated from a (simpler) set of latent generating factors $z$. Assuming the latent factors are independent, the idea of learning *disentangled* representations involves learning these latent factors of variation in an unsupervised manner [15]. However, despite apparent empirical progress in learning disentangled representations [14, 16, 17], there remains inherent issues in both learning and defining disentangled representations [18]. In many cases, different independent latent factors may lead to equivalent observed data, and without an inductive bias, disentanglement remains ill-defined. For example, color can be decomposed into an RGB decomposition, or an equivalent HSV decomposition.

In [18, Theorem 1] it is shown that without any inductive bias, one cannot uniquely identify the underlying independent latent factors in a purely unsupervised manner from observed data. Empirically, they also found that there was no clear correlation between training statistics and disentanglement scores without supervision. Later, and related to our work, the authors examined the setting where there is paired data and no explicit supervision (weak supervision), and found that such a setup was helpful for learning disentangled representations [19]. The authors examined the setting in which the set of shared latent factors changed for each example, which was necessary for their identifiability proof. This also required using the same encoder for each view, and thus is a restricted setting that does not easily scale to multi-modal data.

Here, we study the scenario where the set of generating factors is the same across examples, as in the case of a pair of fixed sensors receiving correlated data. Additionally, our objective is motivated in an information theoretic way and our method generalizes to the case where we have different sensory modalities, which is relevant to neuroscience and multi-modal learning. Finally, our variational objective is flexible and allows estimation of the *common information* in a principled way.

**Approximating Mutual Information**   Estimating mutual information from samples is challenging for high-dimensional random vectors [20]. The primary difficulty in estimating mutual information is constructing high-dimensional probability distribution from samples, as the number of samples required scales exponentially with dimensionality. This is impractical for realistic deep learning tasks where the representations are high dimensional. To estimate mutual information, [21] used a binning approach, discretizing the activations into a finite number of bins. While this approximation is exact in the limit of infinitesimally small bins, in practice, the size of the bin affects the estimator [22, 23]. In contrast to binning, other approaches to estimate mutual information include entropic-based estimators (e.g., [23]) and a nearest neighbours approach [24]. Although mutual information is difficult to estimate, it is an appealing quantity to summarily characterize neural network behavior because of its invariance to smooth and invertible transformations. In this work, rather than estimate the mutual information directly, we study the "usable information" in the network [25, 26], which corresponds to a variational approximation of the mutual information [27, 28].

**Contrastive and Multi-View Approaches**    While (multi-view) contrastive learning aims to learn a representation of *only* the common information between views [7, 29, 30], we aim to learn a decomposition of the information in the views into common and unique components. Our work naturally extends to multi-sensor data that have different amounts of common/unique information (e.g., touch and vision). Moreover, contrastive approaches assume that the unique information is nuisance variability, and discard this information. Similarly, [31] also seeks to identify common information in both views, but also does not provide an objective to retain the unique information. While the multi-view literature is broad, we are not aware of previous attempts to quantify the common and unique information. Related to our approach, [32] aim to find shared and private representations using VAEs, but it differs in how the alignment of shared information is specified and the resulting objective, and they do not provide a way to quantify the information content of the private and shared components. We discuss additional related work in Appendix C.

## 3    Method: Gács-Körner Variational Auto-Encoder

Our formulation involves generalizing the Gács-Körner common information in eq. (1) to the case where $f$ and $g$ are stochastic functions so that the optimization problem becomes:

$$\tilde{C}_{GK}(x_1; x_2) := \max_z I(x_i; z) \tag{4}$$

$$\text{s.t. } z = f_s(x_1) = g_s(x_2), \tag{5}$$

where $f_s$ and $g_s$ are *stochastic* functions. By the equality in eq. (5), we mean that $p(z|x_1) = p(z|x_2)$ for all $(x_1, x_2) \sim p(x_1, x_2)$. Note that when $f$ and $g$ are deterministic functions[3] (which are a subset of stochastic functions), then $H(z|x_i) = 0$ and we recover the original definition since

$$I(x_i; z) = H(z) - H(z|x_i) = H(z). \tag{6}$$

Our latter generalization (eq. 4-5) is more amenable to optimization and interpretable, as we will later demonstrate. In eq. (4), we used $x_i$ as a placeholder since when $p(z|x_1) = p(z|x_2)$ for all $(x_1, x_2) \sim p(x_1, x_2, z)$ then $I(z; x_1) = I(z; x_2)$ since

$$\begin{aligned}
I(z; x_1) &= \mathbb{E}_{x_1 \sim p(x_1)}[KL(p(z|x_1)||p(z))] \\
&= \mathbb{E}_{(x_1, x_2) \sim p(x_1, x_2)}[KL(p(z|x_1)||p(z))] \\
&= \mathbb{E}_{(x_1, x_2) \sim p(x_1, x_2)}[KL(p(z|x_2)||p(z))] \\
&= \mathbb{E}_{x_2 \sim p(x_2)}[KL(p(z|x_2)||p(z))] \\
&= I(z; x_2).
\end{aligned}$$

This means that another equivalent formulation to maximize is $\max_z \frac{1}{2} \sum_i I(x_i; z) = \max_z I(x_i; z)$, for any $i$. To optimize the objective in eq. 4-5, we need to learn a set of latent factors $z$ that maximize $I(x_i; z)$, while satisying the constraint in eq. 5. We propose an optimization reminiscent of the VAE objective. Define $x = (x_1, x_2)$ as the concatenation of both views, and $z = (z_u^1, z_c, z_u^2)$ as a decomposition of the representation into common and unique components, and $z_i = (z_u^i, z_c)$. In particular, we seek to learn latent encodings through an encoder $q_\phi(z|x)$, which maps $x$ to $z$. To optimize the objective, the representation $z$ should maximize $I(x_i; z)$, and so we should also learn a decoder $p_\theta(x|z)$ that minimizes $H(x_i|z)$. This corresponds to the reconstruction term in a traditional VAE, though note here we reconstruct both views.

$$\mathcal{L}_{\text{CVAE}}^1 = \mathbb{E}_{p(x)}[\, \mathbb{E}_{q_\phi(z|x)}[-\log p_\theta(x|z)]] \tag{7}$$

Without any constraints, this could be achieved trivially by using an identity mapping. To ensure that the latents encode only common information between the different views, we decompose the encodings to ensure the following constraint corresponding to eq. (5):

$$D(q_{\phi_{c_1}}, q_{\phi_{c_2}}) = KL(q_{\phi_{c_1}}(z_c|x_1) \,||\, q_{\phi_{c_2}}(z_c|x_2)) = 0. \tag{8}$$

Here $q_{\phi_{c_i}}(z_c|x_i)$ maps $x_i$ to $z_{c_i}$ Rather than enforcing a hard constraint, in practice it is easier to optimize the corresponding Lagrangian relaxation:

$$\mathcal{L}_{\text{CVAE}}^2 = \mathbb{E}_{p(x)}[\, \mathbb{E}_{q_\phi(z|x)}[-\log p_\theta(x|z)] + \lambda_c D(q_{\phi_{c_1}}, q_{\phi_{c_2}})]. \tag{9}$$

---

[3]If the constraint in Eq. 5 is satisfied, the map to the *mean* of the posterior $p(z|x_1)$ (and $p(z|x_2)$) for observations $(x_1, x_2)$ is a deterministic function and satisfies the constraint of Gács-Körner common information.

After optimizing this objective, for a sufficiently large $\lambda$ so that $D(q_1, q_2) \approx 0$, the common information would be:

$$\tilde{C}_{GK}(x_1; x_2) = \mathbb{E}_{p(x)}[\, KL(q_{\phi_{c_i}}(z_c|x_i) \,||\, q^*(z)) \,], \tag{10}$$

where $q^*(z)$ is the marginal distribution induced by the encoder. However, estimating the true marginal $q^*(z)$ is difficult for high-dimensional problems. In practice, we follow [9] and learn an approximate prior $p(z) \approx q^*(z)$, where both $q_\phi(z|x)$ and $p(z)$ are taken from a given family of distributions (such as multivariate Gaussians with diagonal covariance matrix). This will additionally enable us to sample from the distribution, and interpret the latent factors. To learn $p(z)$ we also add the following regularization to our training objective:

$$\mathbb{E}_{p(x)}[\, KL(q_\phi(z|x) \,||\, p(z)) \,]. \tag{11}$$

Alternatively, we can also exploit the degree of freedom in learning $q_\phi(z|x)$ and fix $p(z)$ to be $\mathcal{N}(0, I)$. In both cases, our overall objective becomes:

$$\mathcal{L}_{\text{CVAE}} = \mathbb{E}_{p(x)}[\, \mathbb{E}_{q_\phi(z|x)}[-\log p_\theta(x|z)] + \lambda_c D(q_{\phi_{c_1}}, q_{\phi_{c_2}}) + \beta KL(q_\phi(z|x) \,||\, p(z))]. \tag{12}$$

Optimizing this objective alone could lead to unexplained components of information, for example the unique components. Alternatively, unique information present in the individual views may be encoded in the "common" latent variable if the reconstruction benefits outweighed the cost of the divergence between the posteriors of the encoders (the term corresponding to the $\beta$).

In addition to these common latent components, we can learn unique latent components by optimizing a traditional VAE objective (i.e. with $\lambda_c = 0$) for a subset of the latent variables. Importantly we also need to ensure that the KL penalty for the unique component subset is greater than for the common subset (so that it is beneficial to encode common information in the common latent components). Our final objective becomes

$$\mathcal{L}_{\text{CVAE}} = \mathbb{E}_{p(x)}[\, \mathbb{E}_{q_\phi(z|x)}[-\log p_\theta(x|z)] + \lambda_c D(q_{\phi_{c_1}}, q_{\phi_{c_2}})$$
$$+ \sum_{i=1}^{2} \beta_c KL(q_{\phi_{c_i}}(z_c|x_i) \,||\, p(z_c)) + \beta_u KL(q_{\phi_{u_i}}(z_u|x_i) \,||\, p(z_u))], \tag{13}$$

where $\beta_c$ and $\beta_u$ correspond to a multiplier enforcing the cost of encoding common and unique information respectively. Importantly $\beta_u > \beta_c > 0$, resulting in a larger penalty on the unique latent variables (otherwise all the information would be encoded in the "unique" components). The summation in the bottom part of Eq. 13 corresponds to a summation over both encoders. $p(z_u)$ and $p(z_c)$ are both sampled from $\mathcal{N}(0, I)$ of appropriate dimensionality.

We now show that, if the network architecture used for the VAE implements a generic enough class of encoder/decoders our method will recover the GK common information.

**Theorem 3.1** (GK VAE recovers the common information). Suppose our observations $(x_1, x_2)$ have GK common information defined through the random variable $z_c$ satisfying eq. 4-5 and that our parametric function classes $q(z|x)$ and $p(x|z)$ optimized over can express any function. Then, our optimization (with $\beta_c = 0$, $\beta_u < 1$ and decoder $p(x|z) = \sum_{i=1}^{2} p_i(x_i|z_i)$) will recover latents $\hat{z} = (\hat{z}_u^1, \hat{z}_u^2, \hat{z}_c)$ where $\hat{z}_c$ is the common random variable that maximizes the "stochastic" GK common information in eq. 4-5, while $\hat{z}_u^i$ is the unique information of the $i$-th view, which maximizes $I(x_i; z_u^i, \hat{z}_c)$.

We provide the proof in Appendix A. Note that while the previous theorem guarantees that we will be able to separate the common and unique factors at the block level, we might not be able to disentangle the individual common factors.

## 3.1 Quantifying the common information

Suppose $D(q_{\phi_{c_1}}, q_{\phi_{c_2}}) = 0$. The term corresponding to the rate $R_c$ of the VAE

$$R_c = I_q(z_c; x) = \mathbb{E}_{p(x)} KL(q_{\phi_c}(z_c|x) \,||\, p(z_c)) \tag{14}$$

is neither an upper nor lower bound on the true common information. It represents an upper bound to the information encoded in the representation specified by $q_{\phi_c}(z_c|x)$, but does not bound the true common information in the data, since $q_{\phi_c}(z_c|x)$ itself is a variational approximation.

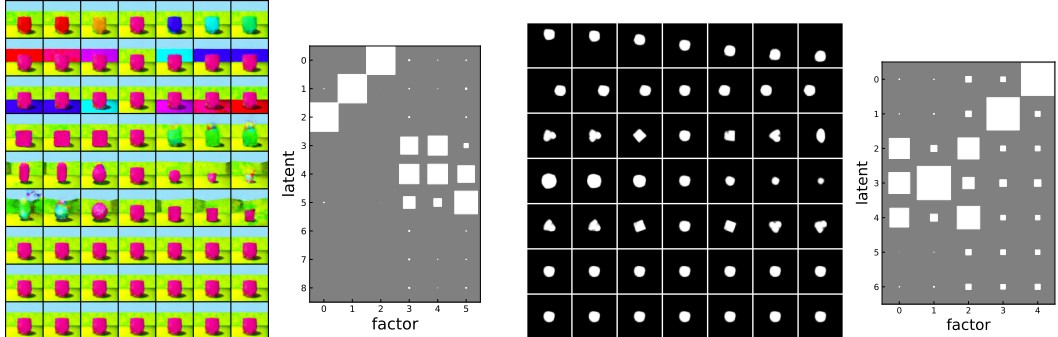

Figure 2: **Latent traversals and DCI plots show optimization results in separation of common and unique information. (Left) 3dshapes:** The top 3 rows shows the unique latents, the middle 3 the common (and the bottom 3 are the unique latents for the second view). The ground truth unique generative factors are $(0, 1, 2)$ corresponding to floor color, wall color, and object color. Our model correctly recovers that those factors are unique (first three rows in the figure), and that the other factors are common (middle three rows). **(Right) dsprites:** The top 2 rows shows the unique latent variables, the middle 3 the common (and the bottom 2 are the unique latent variables for the second view). The ground truth unique generative factors are $(3, 4)$ corresponding to x-position, y-position respectively. Our model correctly recovers that those factors are unique (first two rows in the figure), and that the other factors are common (middle three rows).

To find a lower bound on the common information encoded in the dataset, we can use any mutual information estimator $\hat{I}$ that is a lower bound (see [28] for several). The approximate common information can then be quantified by $\hat{I}(z_q, x)$, where $z_q \sim q_{\phi_c}(z_c|x)$. We report both the rate $R_c$ and $\hat{I}$ in the paper. We emphasize that $\hat{I}$ can be any mutual information estimator. When the data generating distribution is known, as in our synthetic examples, we employ the "Usable Information" estimator, described in Sect. 4.2, which is a variational approximation [27].

## 3.2   Identifiability of the common and unique components

We now show the conditions under which our optimization will identify the common and unique latent components. Usually we do not directly observe the latent factors $z$, but rather an observation generated from them. We may then ask whether the common latent factors can still be reconstructed from this observation. The following proposition shows that this is indeed the case, as long as the function generating $f$ the observation is invertible, i.e., we can recover the latent factors from the observation itself.

**Proposition 3.1.** ([3], Ex. 1): Define

$$z_1 = (z_c, z_u^1), \quad z_2 = (z_c, z_u^2)$$

where $z_c$, $z_u^1$, and $z_u^2$ are mutually independent. Then for any invertible transformation $t_i$ the random variable $z_c$ encodes all the common information:

$$z_c = \arg\max_{\hat{z}} C_{GK}(t_1(z_1), t_2(z_2))$$

We provide the proof in Appendix A. The above proposition shows that when a set of factors is shared between views and when the unique factors are sampled independently, then the GK common random variable corresponds to shared latent factors. In particular, if the observations $x_i$ are generated through an invertible function $x_i = f(z_c, z_u^i)$ where $z_c \sim p(z_c)$ corresponds to the shared factors, the proposition shows that such factors can be recovered from the observations by maximizing the GK common information. In our GK VAE optimization, we optimize the "stochastic" GK common information and we also find in our experiments that we can (approximately) recover the latent factors from observations $x_i$ generated from this process.

|  | FLOOR HUE (10) | WALL HUE (10) | OBJ. HUE (10) | SCALE (8) | SHAPE (4) | ANGLE (15) | KL TOTAL |
|---|---|---|---|---|---|---|---|
| COMMON | -0.01 | -0.02 | -0.03 | 2.73 | 1.98 | 3.83 | 15.0 |
| UNIQUE | 3.31 | 3.31 | 3.31 | 0.19 | 0.37 | 0.19 | 12.7 |
| TOTAL | 3.31 | 3.31 | 3.31 | 2.69 | 1.98 | 3.82 | 27.7 |

Table 1: Usable Information (in bits) in representation for *3dShapes*. The ground truth unique generative factors are *floor color*, *wall color*, and *object color*, and the common generative factors are *scale*, *shape* and *angle*. The common information is separated from the unique information. The ground truth factors were almost perfectly encoded in the latents. The numbers in parenthesis represents the number of discrete factors for each latent variable.

## 4    Experiments

We train our GK-VAE models with Adam using a learning rate of $0.001$, unless otherwise stated. When the number of ground truth latent factors is known, we set the size of the latent vector of the VAE equal to the number of ground truth factors. This choice was not necessary, and we obtain analogous results when the size of the latent vector of the VAE was larger than the number of ground-truth factors (Fig. 12). To improve optimization, we use the idea of free bits [33] and we set $\lambda_{\text{free-bits}} = 0.1$. This was easier than using $\beta$ scheduling [16], since it only involved tuning one parameter. We set $\beta_u$ to be 10, $\beta_c$ to be 0.1 and $\lambda_c = 0.1$. We trained networks for 70 epochs, except for the MNIST experiments, where we trained for 50 (details in the Appendix).

To ensure that the latents are shared to both encoders, during training we randomly sample $z$ from either encoder $q_{\phi_i}(z_c|x_i)$ with $p = 0.5$. We opted to randomly sample the latents from each encoder, as opposed to performing averaging, to ensure that the latent will always be a function of an individual view $x_i$. This is in addition to the soft constraint governed by $\lambda_c$ in the loss.[4]

### 4.1   Evaluation Datasets

We primarily focus on the setting where the ground truth latent factors and generative model are known, in order to quantitatively benchmark our approach. To do so, we constructed datasets with ground truth latent factors so that some of the latent factors are shared between each views. That is, the generative model for the data $(x_1, x_2)$ is

$$x_1 = f(z_u^1, z_c), \quad x_2 = f(z_u^2, z_c), \tag{15}$$

where $z_c$ is shared between the views and $z_u^i$ is the unique information encoded in the $i^{th}$ view and $f$ corresponds to a rendering function.

To construct such datasets, we modified the *3dshapes* [6] and the *dsprites* dataset [5]. We select a subset of the latent factors to be shared between the views, while the remaining factors are sampled independently for each view. The *3dshapes* dataset [6] contains six independent generating factors: floor color, background color, shape color, size, shape, and viewpoint. Each latent factor can only take one of a *discrete* number of values. The *dsprites* dataset [5] contains five independent generating factors: shape, scale, rotation, x and y position. Each latent factor can only take one of a *discrete* number of values. When we generate multi-view data following the generative model in eq. (15), we refer to these datasets as *Common-3dshapes* and *Common-dsprites* respectively. We consider additional variants in the Appendix.

We also examine the *Rotated Mnist Dataset*. where the two views are two random digits of the same class to which a random rotation is applied. In particular, the class of the digit is common information between the views whereas the rotation is unique. We also examine the synthetic video dataset *Sprites* (not to be confused with *dsprites*) described in [34] and evaluate the common information in frames separated $t$ frames apart.

### 4.2   Metrics

**DCI Disentanglement Matrix [35].**    Let $d$ be the dimension of the latent space and let $t$ be the true generating factors. The idea is to train a regressor $f_j(z) : \mathbb{R}^d \to \mathbb{R}$ to predict the ground truth factors

---

[4]Our experiments can be reproduced in approximately 3 days on a single GPU (g4dn instance).

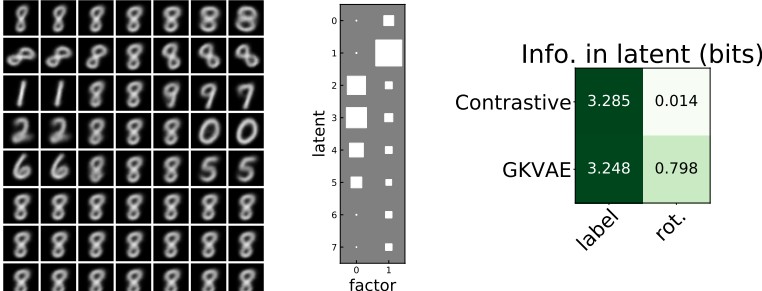

Figure 3: **Left.** Traversals for *Rotated Mnist*. The unique components of the latent (rows 1,2) appear to encode the "thickness" and rotation of the digit, whereas the common components appear to represent the overall digit (rows 3-6); and also the output of view 1 does not depend on the latents in rows 7,8 (these correspond to the unique components for view 2.) **Middle.** Corresponding DCI matrix, where factor 0 corresponds to the label, while factor 1 corresponds to the rotation (discretized into 10 bins). **Right.** Comparison against contrastive implementation from [7], where the contrastive approach does not encode any usable information about the unique factor (the rotation).

$t_j$ for each $j$. This results in a matrix of coefficients that describe the importance of each latent for predicting each ground truth factors. This can then be visualized as a matrix where the size of the square reflects the coefficient. We use this metric to assess the partitioning of the learned common and unique representations. We used the random forest regressor, similar to [18] to predict a *discrete* number of latent classes.

**Usable Information [25, 36, 26].** We use this to approximate the mutual information when $H(x)$ is known, as it is in the datasets previously described. It is a lower bound to mutual information. We use this to lower bound the information contained in the representation $z$ in the next section.

### 4.3 Results

**Separation of Common and Unique Latent Variables.** We first examine whether our formulation can correctly separate the common and unique latent factors. After optimizing a network on our *Common-3dshapes* dataset we examined how much information about the ground-truth latent factors were encoded in the common latents $z_c$ and the unique latents $z_u$ (Table 1).

Given the encoded representation specified by $q_\phi(z|x)$, we evaluated the usable information for the two latent components ($z_c$ and $z_u$), as well as by using the complete latent variable $z$. As done in previous work [18], we directly use the mean of $q_\phi(z|x)$ as our representation $z$ rather than sampling. In Table 1, we see that the common and unique information was perfectly separated. Note, that information values reported are a lower bound to the true information, as our variational approximation is a lower bound to $I_q(z;x)$ (which is itself a variational approximation). Our method accurately encodes all common information between views (ground truth: 3.32 bits for floor, wall, and background hue; 3 bits for scale; 2 bits for shape; 3.91 bits for orientation).

We also performed these analyses on the *Common-dsprites* dataset and found similar results (Table 2, Appendix). In particular, the unique latent factors corresponding to position are encoded in the unique components of the latent representation, while the other factors are encoded in the common latent representation. We emphasize that the generative model was not used at all during training, and was only used for quantitative evaluation after training.

In Fig. 2, we show the DCI matrix [35] which visually reaffirm that the common and unique factors are properly identified at the block-level for both the *Common-3dshapes* and *Common-dsprites* datasets. We also include traversals of the prior shown in Fig. 2 to show qualitatively that the learned factors of variation are meaningful and can be interpreted. Additional runs are in Appendix F.

**Rotated Mnist and Comparison with Contrastive Learning.** As described before, we generate a dataset of paired views of digits of the same class, each rotated by an independent random amount. In this manner, the unique information is about the rotation, whereas the common information is about the class. In Fig. 3 we see that the unique components of the latent (rows 1, 2) appear to encode the

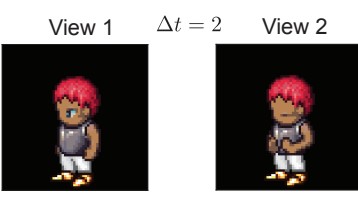 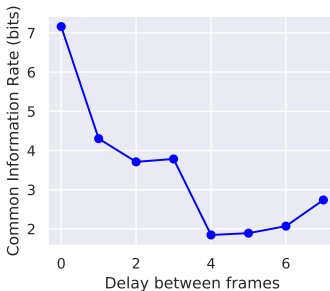

View 1    $\Delta t = 2$    View 2

Figure 4: Sprites [34] video experiment. **(Left)** Example views separated 2 frames apart. **(Right)** Common information as a function of delay between frames. In general common information is decreasing as the delay gets longer.

rotation and "thickness" of the digit, whereas the common components seem to represent the class of the digit (rows 3-6). Also, as expected, the output of view 1 does not depend on the latents in rows 7, 8 which by construction correspond to the unique components of view 2.

This setup is reminiscent of contrastive learning, where the goal is to learn a representation which is invariant to a random data augmentation of the input (such as a random rotation). By construction, contrastive learning aims to encode the common information before and after data augmentation, but may not encode any other information. This can lead to degraded performance on downstream tasks, as the discarded unique information may still be important for the task [29, 37]. On the other hand, our GK-VAE separates the unique and common information without discarding information.

To highlight this difference between approaches, we trained using a contrastive objective[5] [7], and found that indeed while we can decode the shared class label, we cannot decode the unique rotation angle of view 1 (discretized into 10 bins; Fig. 3, right). On the other hand, using our method we recover the common and unique information.

**Common information across time in sequences from videos.** The existence of common information though time is another important learning signal. To study it, we perform an experiment on the *Sprites* dataset described in [34]. This dataset consists of synthetic sequences all with 8 frames. We optimized using the same architecture and hyperparameters except we set $\lambda_c = 0.5$. We examine the common information between frames $t$ frames apart, approximated using the KL divergence term. In particular, the two views are two frames $(x_1, x_{t+1})$, where each pair belongs to a different video sequences. In Fig. 4 we see that in general, as $t$ increases the common information between the frames decreases evidencing the fact that, due to the random temporal evolution of the video, common information is lost as time progresses. We also note that the common information appears to increase in the last frame; this could be that in many of the sequences the sprite returns close to the initial state (see Fig. 3 in [34]).

## 5    Discussion

We show formally and empirically that we can partition the latent representation of multi-view data into a common and unique component, and also provide a tractable approximation for the Gács-Körner common information between high dimensional random variables, which has been a difficult problem [4]. In many practical scenarios where high dimensional data comes from multiple sensors, such as neuroscience and robotics, it is desirable to understand and quantify what is common and what is unique between the observations. Motivated by the definition of common information proposed by Gács and Körner [2], we propose a variational relaxation and show that it can be efficiently learned from data by training a slightly modified VAE. Empirically, we demonstrate that our formulation allows us to learn semantically meaningful common and unique factors of variation. Our formulation is also a generative multi-view model that allows sampling and manipulation of the common and unique factors.

---

[5]We used the code from: https://github.com/HobbitLong/CMC (BSD 2-Clause License)

As the common information was motivated by an information theoretic coding problem [2], our work naturally relates to compression schemes. Indeed, approximate forms of the common information, discussed further in Appendix C, are scenarios for distributed compression, since the common information needs to only be transmitted once [38, 4]. It may be interesting to combine our approach with recent advances in practical compression algorithms that leverage VAEs [39].

## Acknowledgments

MK was supported by the National Sciences and Engineering Research Council (NSERC) and a fellowship from the Science Hub for Humanity and Artificial Intelligence (UCLA/Amazon). JCK acknowledges financial support from NSF CAREER 1943467. SS acknowledges financial support from ONR N00014-22-1-2252.

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

# A  Proofs

**Theorem A.1** (GK VAE recovers the common information). Suppose our observations $(x_1, x_2)$ have GK common information defined through the random variable $z_c$ satisfying eq. 4-5 and that our parametric function classes $q(z|x)$ and $p(x|z)$ optimized over can express any function. Then, our optimization (with $\beta_c = 0$, $\beta_u < 1$ and decoder $p(x|z) = \sum_{i=1}^{2} p_i(x_i|z_i)$) will recover latents $\hat{z} = (\hat{z}_u^1, \hat{z}_u^2, \hat{z}_c)$ where $\hat{z}_c$ is the common random variable that maximizes the "stochastic" GK common information in eq. 4-5, while $\hat{z}_u^i$ is the unique information of the $i$-th view, which maximizes $I(x_i; z_u^i, \hat{z}_c)$.

*Proof.* We consider the hard-constrained optimization problem with an infinitely expressive function class (i.e. so that the cross-entropy loss corresponds to the conditional entropy). Our VAE objective corresponds to the optimization problem

$$\min_{z_c, z_u^1, z_u^2} \quad H(x_1|z_u^1, z_c) + \beta_u I(z_u^1; x) + \beta_c I(z_c; x) +$$
$$H(x_2|z_u^2, z_c) + \beta_u I(z_u^2; x) + \beta_c I(z_c; x)$$
$$\text{s.t.} \quad D(q_{\phi_1}, q_{\phi_2}) = 0$$

We consider a sequential optimization of finding $\hat{z}_c$ and $\hat{z}_u^i$, and then show that this solution minimizes the joint objective above. We first consider the hard-constrained version of eq. (12).

$$\min_{z_c} \quad H(x_1|z_c) + \beta_c I(z_c; x_1) +$$
$$H(x_2|z_c) + \beta_c I(z_c; x_2)$$
$$\text{s.t.} \quad D(q_{\phi_1}, q_{\phi_2}) = 0$$

Note that $H(x_i) = H(x_i|z_c) + I(z_c; x_i)$. We can rewrite the loss as:

$$\mathcal{L} = H(x_1|\hat{z}_c) + H(x_2|\hat{z}_c) + \beta_c(I(\hat{z}_c, x_1) + I(\hat{z}_c, x_2))$$
$$= H(x_1) + H(x_2) + (\beta_c - 1)(I(\hat{z}_c, x_1) + I(\hat{z}_c, x_2))$$

This tells us that the optimal $z_c$ maximizes $I(\hat{z}_c, x_1) + I(\hat{z}_c, x_2)$. This is exactly the definition that we give of "stochastic" GK common information. Note we have previously shown $I(z_c; x_1) = I(z_c; x_2)$. Given $\hat{z}_c$ found above, the remaining objective (eq. (13)) becomes:

$$\min_{z_u^1, z_u^2} \quad H(x_1|z_u^1, \hat{z}_c) + \beta_u I(z_u^1; x_1) +$$
$$H(x_2|z_u^2, \hat{z}_c) + \beta_u I(z_u^2; x_2).$$

For $\beta_u < 1$, the objective maximizes $I(x_1; z_u^1, \hat{z}_c)$, which was the definition of the unique information. (For $\beta_u > 1$, this corresponds to a $\beta$-VAE, and will have the corresponding trade-off between rate and reconstruction [14, 11, 12].)

Finally, suppose $\tilde{z}_c$ did not contain all the common information as $\hat{z}_c$; i.e. $I(\tilde{z}_c; x_i) < I(\hat{z}_c; x_i)$. Write the final equation as a maximization by noting that

$$H(x_i|z_u^i, \hat{z}_c) = -I(x_i; z_u^i, \hat{z}_c) + H(x_i)$$

Then the final optimization (for any $i$) is equivalent to

$$\max_{z_u^i} I(x_i; z_u^i, \hat{z}_c) - \beta_u I(z_u^i; x_i) - H(x_i) = \max_{z_u^i} I(x_i; \hat{z}_c) + I(x_i; z_u^i|\hat{z}_c) - \beta_u I(z_u^i; x_i) - H(x_i)$$
$$(16)$$

$$> \max_{z_u^i} I(x_i; \tilde{z}_c) + I(x_i; z_u^i|\tilde{z}_c) - \beta_u I(z_u^i; x_i) - H(x_i)$$
$$(17)$$

$$= \max_{z_u^i} I(x_i; z_u^i, \tilde{z}_c) - \beta_u I(z_u^i; x_i) - H(x_i) \qquad (18)$$

For any $z_u^i$, Eq. 16 is maximized with $\hat{z}_c$ that encodes all the common information. Thus the GK VAE optimization is minimized with $\hat{z} = (\hat{z}_u^1, \hat{z}_u^2, \hat{z}_c)$. □

**Proposition A.1.** ([3], Ex. 1): Define
$$z_1 = (z_c, z_u^1), \quad z_2 = (z_c, z_u^2)$$
where $z_c, z_u^1$, and $z_u^2$ are mutually independent. Then for any invertible transformation $t_i$ the random variable $z^*$ that satisfies
$$\arg \max_{\hat{z}} C_{GK}(t_1(z_1), t_2(z_2))$$
is $z_c$.

*Proof.* Note that if $t$ is the identity transformation $t(z) = z$, then
$$\arg \max_{\hat{z}} C_{GK}(z_1, z_2)$$
is $z_c$. In general, if $t_i$ are invertible transformations, suppose that $f_1$ and $f_2$ are the functions satisfying $\hat{z} = f_1(z_1) = f_2(z_2)$ corresponding to $C_{GK}(z_1, z_2)$. Then the functions corresponding to $C_{GK}(t_1(z_1), t_2(z_2))$ will be $\hat{z} = f_1 \circ t_1^{-1}(t_1(z_1)) = f_2 \circ t_2^{-1}(t_2(z_2))$ and the random variable $\hat{z}$ is equivalent.

$\square$

# B  Experimental Details

We trained networks with Adam with a learning rate of $0.001$, unless otherwise stated. When the number of ground truth latent factors is known, we set the number of latents equal to the number of ground truth factors. To improve optimization, we use the idea of free bits [33] and we set $\lambda_{free-bits} = 0.1$. This was easier than using $\beta$ scheduling, since it only involved one parameter $\lambda_{free-bits}$. We set $\beta_u$ to be 10 and $\beta_c$ to be 0.1. We trained networks for 70 epochs, except for the Mnist experiments, where we trained for 50 epochs. We used a batch size of 128 and we set $\lambda_c = 0.1$. For all our experiments we used the same encoders and decoders as [16], which has been also used in recent work [18]. We used Gaussian encoders and Bernoulli decoders, basing our implementation off [40]. Our architecture is schematized in Fig. 1. Note that we optimized encoders and decoders separately between the views (i.e weights were not shared).

To ensure that the latents are shared to both encoders, during training we randomly sample $z$ from either encoder $q_{\phi_i}(z_c|x_i)$ with $p = 0.5$. We opted to randomly sample the latents from each encoder, as opposed to performing averaging, to ensure that the latent will always be a function of an individual view $x_i$. This is in addition to the soft constraint governed by $\lambda_c$ in the loss.

In our implementation, unless otherwise noted, we used two separate view-dependent decoders that each processed all the latents $z$ to reconstruct the observations. We also verified that view specific decoders that processed only view-specific latents $z_i$, as in Thm. 3.1, also led to a partitioning of the common and unique information Fig. 5. This was the case both when the unique factors were pairwise independent, as in generative model described in Sect. 4.1 used for the majority of our experiments, and also if there were correlations between the unique components (Fig. 7) and (Fig. 6), with the experiments described in Sect. F. Empirically, we found the separation of the common and unique information was robust to the partitioning of the latents fed to the decoder of our GK-VAE. We provide both decoder implementations in our repository available at https://github.com/mjkleinman/common-vae.

To quantify the information contained in the representation, we calculate the usable information (in bits), which is a lower bound to the information contained in the representation [25, 26]. To train our decoder, we used the `GradientBoostingClassifier` from `sklearn` with default parameters. We trained on 8000 samples and tested on 2000. We evaluated the information on a held-out test set, and hence the negative values correspond to overfitting on the training set. In Table 1, the numbers in parentheses correspond to the number of ground truth factors. We used the same setup for the *rotated Mnist* experiments (Fig. 3). For the rotation angle, we discretized the angle of rotation $(-45°, 45°)$ into 10 bins of equal size, and predicted the discrete bin. We predicted the rotation angle applied to the first view. When comparing with a constrastive learning approach (Fig. 3, right), we used the same encoder backbone as our GK-VAE[6]. We pre-trained with a batch size of 256 for 60 epochs with a learning rate of 0.001, and then trained the linear classifier for 10 epochs with an initial learning rate of 0.03. We used a latent dimension of 20.

---

[6]We used the code from: https://github.com/HobbitLong/CMC

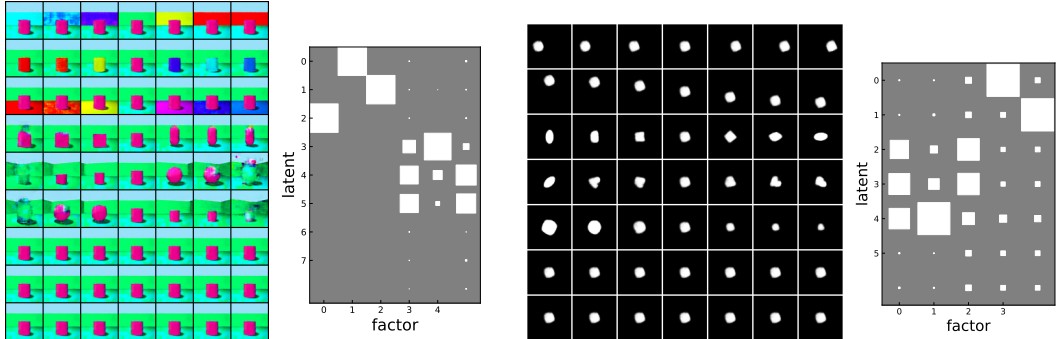

Figure 5: **Latent traversals and DCI plots show optimization results in separation of common and unique information when using partitioned decoder.** Same conventions as Fig. 2. **(Left) 3dshapes:** The top 3 rows shows the unique latents, the middle 3 the common (and the bottom 3 are the unique latents for the second view). The ground truth unique generative factors are $(0, 1, 2)$ corresponding to floor color, wall color, and object color. Our model correctly recovers that those factors are unique (first three rows in the figure), and that the other factors are common (middle three rows). **(Right) dsprites:** The top 2 rows shows the unique latent variables, the middle 3 the common (and the bottom 2 are the unique latent variables for the second view). The ground truth unique generative factors are $(3, 4)$ corresponding to x-position, y-position respectively. Our model correctly recovers that those factors are unique (first two rows in the figure), and that the other factors are common (middle three rows).

## B.1    DCI Plots and Latent Traversals:

Let $d$ be the dimension of the representation $z$ and let $t$ be the true generating factors. The idea is to train a regressor $f_j(z) : \mathbb{R}^d \to \mathbb{R}$ to predict the ground truth factors $t_j$ for each $j$ from the representation $z$. This results in a matrix of coefficients that describe the importance of each component of the representation for predicting each ground truth factors. This matrix $R$ is the *importance matrix* that we visualize in the paper, where $R_{ij}$ reflects the relative importance of of $z_i$ for predicting $t_j$ (where $z_i$ refer to the $i^{th}$ component of the representation). We used the `GradientBoostingClassifier` from `sklearn` with default parameters, similar to [18] to predict the ground truth factors.

We visualize the latent components by traversing one latent variable at a time, while keeping the others fixed. We plot traversals of the prior $p(z)$ and the posterior $q(z|x)$ in different figures (traversals of the prior are in Fig. 2, Fig. 3, and Fig. 5, while the others are posterior traversals). We obtain the posterior traversals by encoding an observation through $q(z|x)$ and traversing each component of the representation.

## C    Other Related Work

A similar formulation has been used for multi-view learning [41], with two separate autoencoders, with an constraint that each latent representation be similar, with the similarity measured by the canonical correlation of the latent representations. They did not motivate it from an information-theoretic perspective; and rather empirically found that such an optimization lead to good representations in the multi-view setting.

Also related to our work is [4]. [4] defined the approximate Gács-Körner information in the following manner:

$$\begin{aligned} \max_{z} \quad & I(x_1; z) \\ \text{s.t.} \quad & H(z|x_2) < \delta \\ & z \leftrightarrow x_1 \leftrightarrow x_2 \end{aligned} \qquad (19)$$

By showing that they could perform the optimization over deterministic functions $f$ such that $z = f(x_1)$, they formed a Lagrangian corresponding to:

$$\max_f \quad H(f(x_1)) - \lambda H(f(x_1)|x_2) \tag{20}$$

They noted that the above optimization is difficult to perform and that future work should look into avenues for computing this quantity; indeed it looks difficult to learn the function $f$ from the above optimization problem. They also suggested that this approximate form of the Gács-Körner common information had potential applications in terms of compression, since the (approximate) common information only needs to be represented once.

### C.1  Relationship to redundant information in the Partial Information Decomposition

Our approach also relates to approaches that aim understand how the information that a set of sources contain about a target variable is distributed among the sources. In particular, [42] proposed the Partial Information Decomposition (PID), which decomposes the information that two sources $X_1$ and $X_2$ contain about a target variable $Y$ into a the components that are *unique*, *redundant*, and *synergistic*. A central quantity in this decomposition, the *redundant information*, reflects the shared information about a *target* variable. The Gacs-Korner common information is equivalent to existing definitions redundant information if the target is reconstructing the sources (i.e, $Y = (X_1, X_2)$) [43]. We note that computing the redundant information from high dimensional samples has been challenging. Recently [44] proposed an approach that could be applied on high dimensional *sources* but where the *target* was low dimensional. Here, our approximation of the common information reflects a further step which can be applied on high dimensional samples (and targets).

### C.2  Other multi-view representation learning approaches for partitioning common/unique information

Our work relates to a growing body of multi-view representation learning approaches aiming to disentangle common and unique information from grouped data. [45] examine the setting of extracting a common content and variable style from a set of grouped images based on content. [46, 47, 48] are also similar in spirit, aiming to disentangle the shared and unique component between paired data.

These existing methods appear to focus on qualitatively partitioning the information through the use of different objectives, however these approaches lack a well-defined notion of common (and unique) information, making it difficult to prove theorectical guarantees for the recovery of the common/unique component, or enable quantification of the information contained in the representation. In contrast, our setup allows us to formalize the decomposition of information using a well-defined information theoretic notion of common/unique information and we show both formally and empirically that we can optimize the objective with a simple modification to a traditional VAE setup and that we can quantify the amount of information contained in the representation.

## D  Limitations of our approach

To validate our approach, we focused on the simpler setting where we have paired data, however, we could extend our formulation to find common information between $n > 2$ sources, as well as finding common information between subsets of sources. While our approach can be naturally extended to find the common information between $n$ sources, future work could investigate a scalable approach to identify common and unique information between arbitrary subsets of the sources. Additionally, to validate our approach empirically, we focused on using a convolutional encoder on relatively small images and video frames, but our formulation is general and the encoder could be interchanged depending on the complexity and inductive biases of the task and data.

## E  Differences between GK Common Information and Mutual Information

The following examples highlight the difference between GK information and mutual information, and suggest why using GK information may be desirable in practical settings.

**Case 1:** Let $X_1 = C + N_1$ and $X_2 = C + N_2$, where $C$ and $N_i$ are independent variables, but the noise $N_1$ and $N_2$ are correlated. Suppose C is discrete – for example, a 0 or 1 feature – and the variance of the noise $N_i$ is relatively small ($\|\Sigma_1\| \ll 1$). In this case, since $C$ can be recovered deterministically from $C + N_i$ (for example by thresholding its value) the GK common information between $X_1$ and $X_2$ is 1 bit. On the other hand, the mutual information between $X_1$ and $X_2$ – which is given by $I(X_1, X_2) = 1 + 0.5\log(\frac{\det \Sigma_1 \det \Sigma_2}{\det \Sigma})$ – can be made arbitrarily large by increasing the correlation between $N_1$ and $N_2$, i.e., making $\det \Sigma \to 0$. Hence, the the value of MI is unrelated to the actual semantic information shared between the two variables. In contrast, the GK common information only identifies the common component $C$. In a sense, the GK is able to recover the underlying "discrete" or "symbolic" information that is common between two different continuous sources. This behavior is valuable in settings like neuroscience, where one wants to measure what is robustly encoded by both representations and not just how much they are correlated (which could be due to any amount of spurious factors). Additionally, with our relaxation of the problem we can measure how much "almost" discrete information is encoded in both, which makes the method more applicable to real-world cases.

**Case 2:** Let $X_1 = f_1(C; U_1)$ and $X_2 = f_2(C; U_2)$ where $C$ and $U_i$ are independent but $U_1$ and $U_2$ are correlated. Suppose $f_1$ is an invertible data generating function mapping latent factors to observations (such as pixels). By maximizing mutual information between views, one implicitly measures those correlations (based on a similar computation to above in Case 1), whereas in the GK sense one cares about only the fully shared components between views. These different perspectives can offer different utility depending on the use case.

We show in Fig. 6 and Fig. 7 that we can recover the partitioning of common and unique factors in practice from the generative model in Case 2.

## F   Additional Experiments

In Table 2, we compute the information encoded in the common and unique latent components for the *common-dsprites* experiment described in the main text, with corresponding DCI matrix and latent traversals in Fig. 2. We also report additional runs for the *common-dsprites* and *common-3dshapes* experiments in Fig. 8 and Fig. 9 respectively. These additional runs are consistent with what was reported in the text, separating the common and unique factors.

To introduce correlations between the unique components (Case 2 in App. E) we modified the generative model from Eq. 15 where the generative model for the data $(x_1, x_2)$ was

$$x_1 = f(z_u^1, z_c), \quad x_2 = f(z_u^2, z_c), \tag{21}$$

where $z_c$ is shared between the views and $z_u^i$ is the unique information encoded in the $i^{th}$ view and $f$ corresponds to a rendering function. To allow the unique components to be dependent (but not the same as the common latent component $z_c$), we modified the generative model so that $z_u^2$ was conditionally dependent on the value of $z_u^1$. We implemented this constraint component-wise for each component of $z_u^i$ by reducing the space of possible latent values of the corresponding components of $z_u^2$ (to half, floored if odd) depending on the component-wise values of $z_u^1$. We show in Fig. 6 and Fig. 7 that we can recover the partitioning of common and unique factors in practice from this generative model.

In addition to the experiments described in the main text, we report variants of *common-dsprites* and *common-3dshapes*. In particular, we change the set of ground-truth common and latent factors.

For the *common-3dshapes*, we specified that the viewpoint was the unique latent variable $z_u$, whereas the other latent variables (background color, floor color, object color, shape, size) were common to both views. We show the DCI matrix and the traversals in Fig 10. For the *common-dsprites* variant, we set the unique components to be the size, scale, and orientation, and the common latent factors to be the x and y position. We show the DCI matrix and latent traversals in Fig. 11. The common and unique latent variables from our optimization separated these ground truth factors.

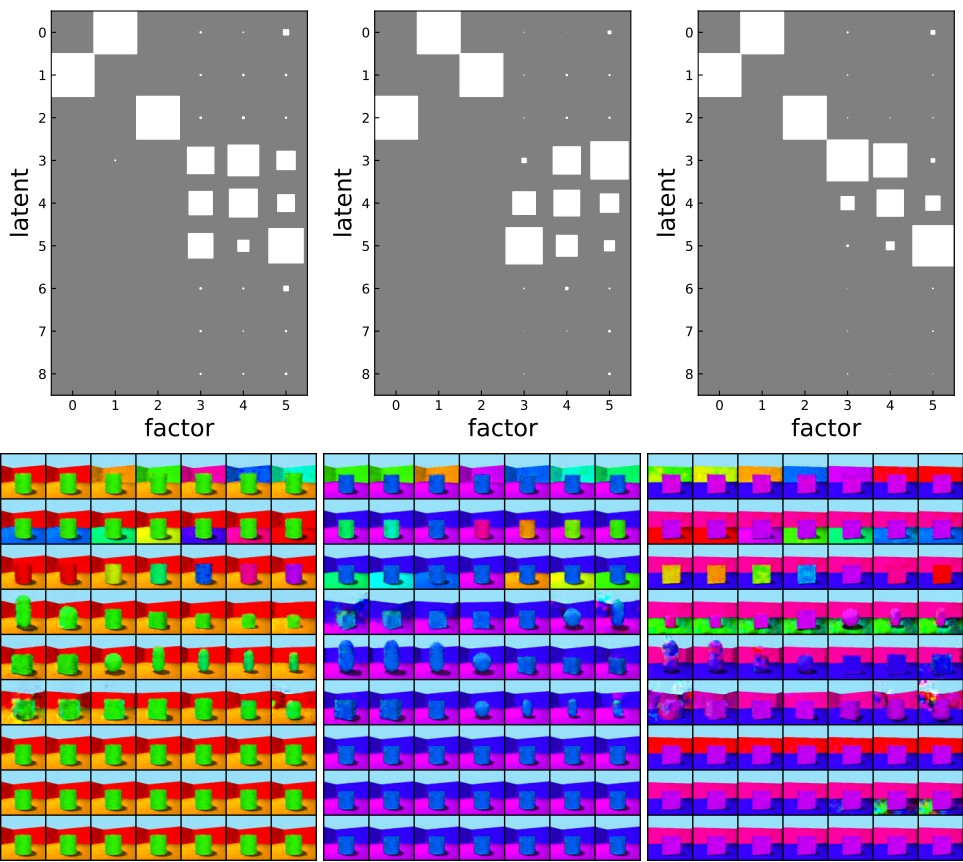

Figure 6: **Decoder from *all latents* partitions common and unique information even when unique generating factors are dependent**. We modified the generative model from Eq. 15 so that the common and unique generating variables were independent but there were correlations between the unique components. In particular, the latent factors for the second view were conditionally dependent on the unique latent factors. We still observed a partitioning of the learned common and unique latent variables, as in the paper. The top 3 rows shows the unique factors, the middle 3 the common (and the bottom 3 are the unique factors for the second view). The ground truth unique generative factors are $(0, 1, 2)$ corresponding to floor color, wall color, and object color. Our model correctly recovers that those factors are unique (first three rows in the figure), and that the other factors are common (middle three rows).

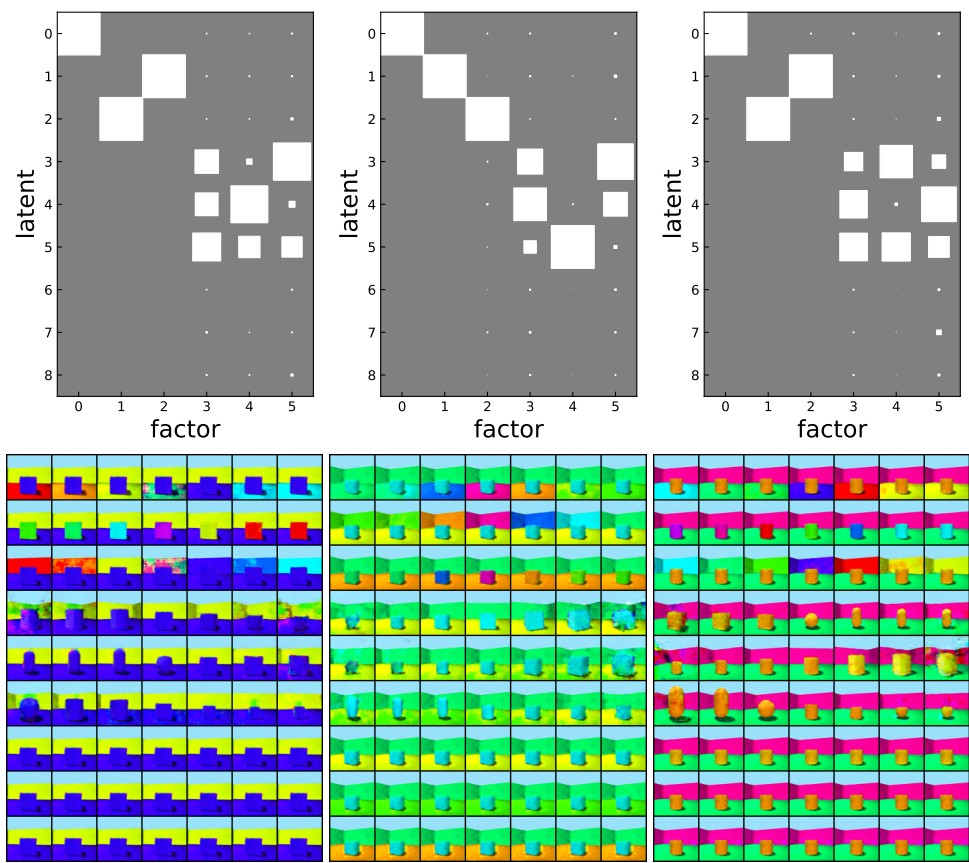

Figure 7: **Decoder from *partitioned latents* also partitions common and unique information, similar to when decoding from all latents in Fig. 6**. The top 3 rows shows the unique factors, the middle 3 the common (and the bottom 3 are the unique factors for the second view). The ground truth unique generative factors are $(0, 1, 2)$ corresponding to floor color, wall color, and object color. Our model correctly recovers that those factors are unique (first three rows in the figure), and that the other factors are common (middle three rows).

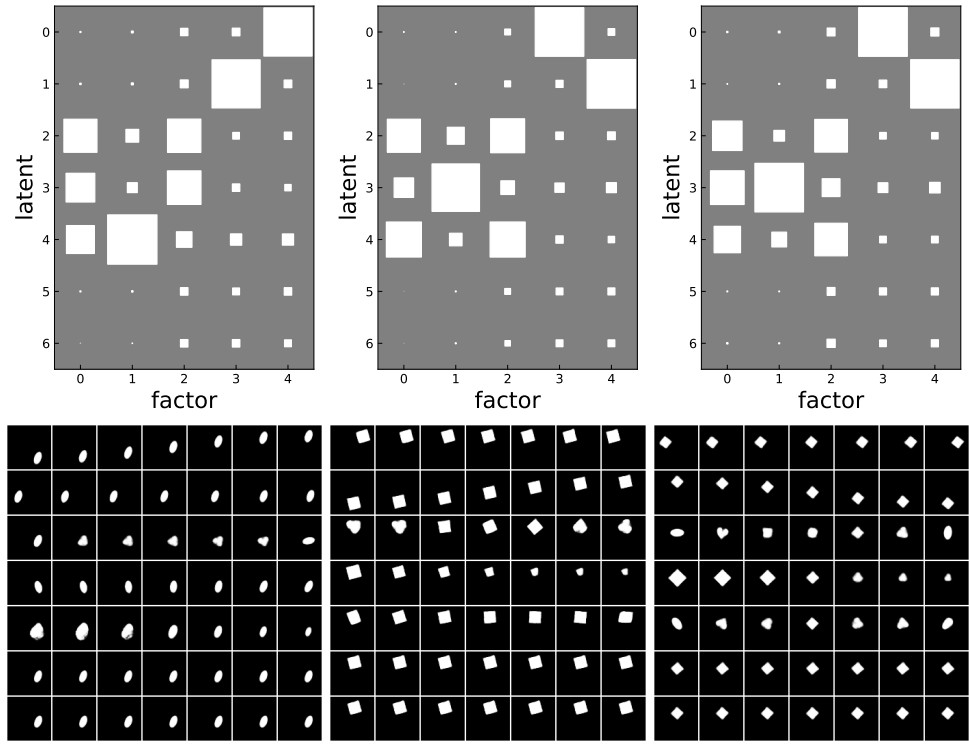

Figure 8: Additional runs for the same experiment as Fig. 2 (right) for *common-dsprites*, with the same conventions as Fig 2. These runs also show the same blockwise partitioning of common and unique information, as in the paper.

| | SHAPE (3) | SCALE (6) | ANGLE (40) | X-POS (32) | Y-POS(32) | KL TOTAL |
|---|---|---|---|---|---|---|
| COMMON | 1.54 | 2.45 | 2.88 | -0.33 | -0.29 | 9.57 |
| UNIQUE | 0.08 | 0.03 | -0.5 | 3.58 | 3.63 | 9 |
| TOTAL | 1.54 | 2.45 | 2.76 | 3.68 | 3.69 | 18.57 |

Table 2: Usable Information (in bits) in representation for a *dsprite* experiment. The common information is separated from the unique information. The ground truth unique factors (x-position and y position) were almost perfectly encoded in the corresponding latents latents, and the other factors (shape, scale, and angle) were correctly encoded in the common latent variables. The numbers in parenthesis represents the number of discrete factors for each latent variable.

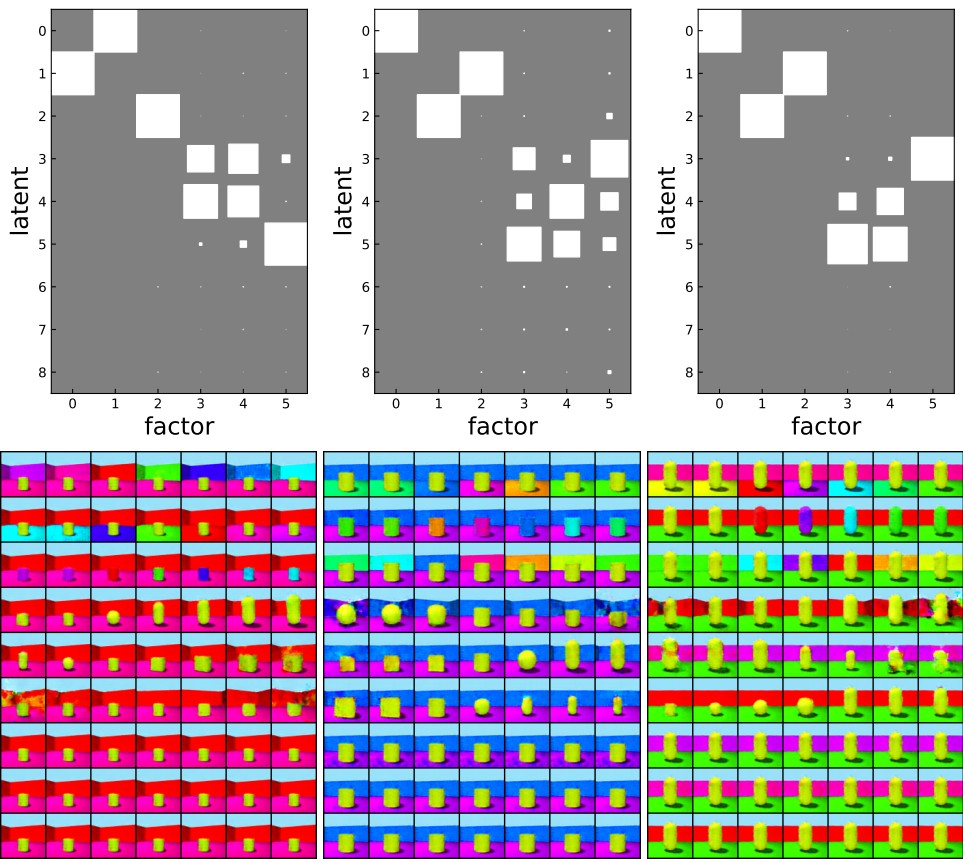

Figure 9: Additional runs for the same experiment as Fig. 2 (left) for *common-3dshapes*, with the same conventions. These runs also show the same blockwise partitioning of common and unique information, as in the paper.

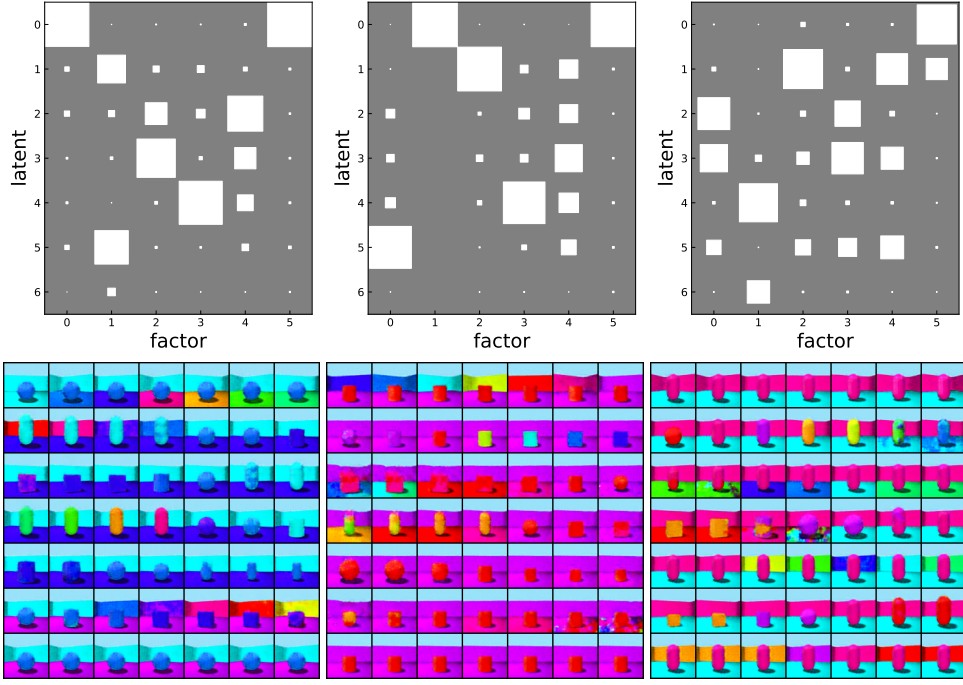

Figure 10: DCI matrix of *common-3dshapes* variant for different random seeds. The top row shows the unique factor, the middle 5 the common (and the bottom row is the unique factors for the second view). The ground truth unique generative factors are (5) corresponding to the viewpoint. Our model generally correctly recovers that the factor is unique (first row in the figure), and that the other factors are common (middle five rows).

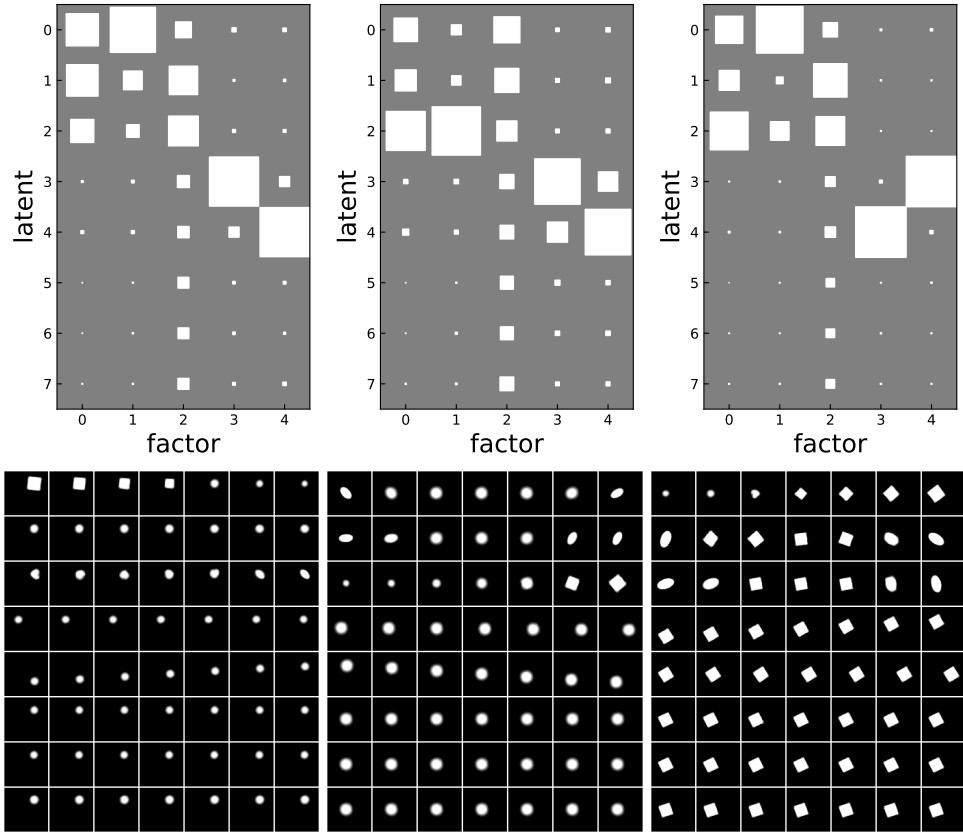

Figure 11: DCI Matrix of *dsprites* experiment variant for different random seeds. The top 3 rows shows the unique latent variables, the middle 2 the common (and the bottom 3 are the unique latent variables for the second view). The ground truth unique generative factors are $(0, 1, 2)$ corresponding to shape, scale, and viewpoint angle. Our model correctly recovers that those factors are unique (first three rows in the figure), and that the other factors (x and y position) are common (middle two rows).

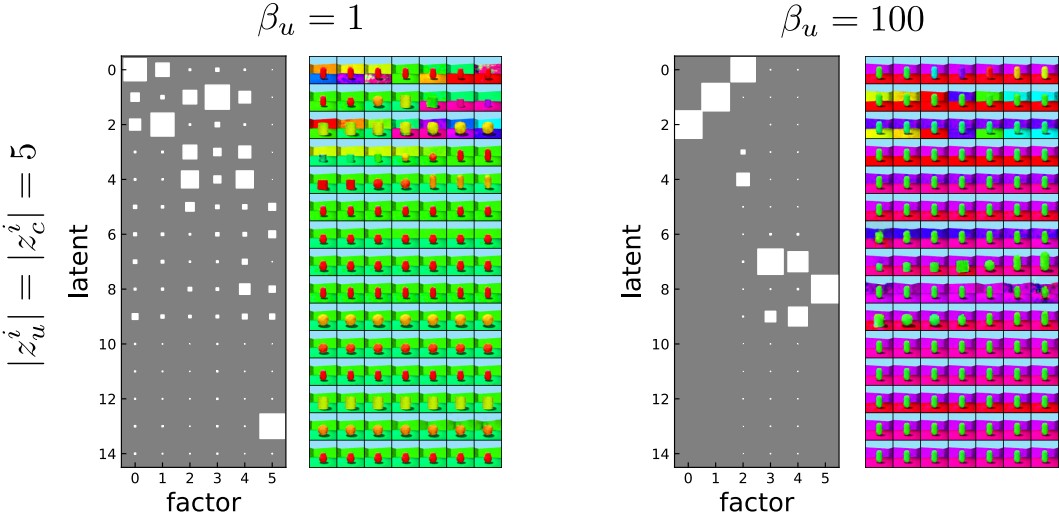

Figure 12: **Effect of** $\beta_u$. Additional runs for Fig. 2 (right) for *common-3dshapes* with 5 unique and 5 common latents per view. For large $\beta_u$ (e.g. $\beta_u = 100$, right), as in the paper, we see a similar blockwise separation of the common and unique latents, even when the number of latents does not match the ground-truth number of latents.

