# OpenReview forum: "Gacs-Korner Common Information Variational Autoencoder"
_NeurIPS.cc/2023/Conference — NeurIPS 2023 poster_

### Official Review · Reviewer_cgd1 · 2023-06-30

**Soundness:** 4 excellent
**Presentation:** 4 excellent
**Contribution:** 3 good
**Rating:** 7
**Confidence:** 3

**Summary:**

The authors present a concept of common information that assesses and distinguishes shared information from unique information between two random variables. This notion, which is a variant of the Gacs-Korner common knowledge, is more optimizable and experimentally approximable when employing sample data.





**Strengths:**

 I believe this paper made a good contributions by proposing to relax the requirement that the representation must be a deterministic function and instead allows it to be a stochastic mapping. Based on the results, it has been shown that such an approach is more  beneficial for quantifying and interpreting the latent representation.





**Weaknesses:**


-  The authors experimentally demonstrating that adding unsupervised viewpoints improves disentanglement; which is likely due to limitations in single-sample analysis; active interaction, not passive observation, fosters better learning of environment representations. but the observation has not been sufficiently explained or explored in the current study.

**Questions:**

- Could you describe how the multi-view VAE model's unique latent variables capture the individual factors that cause variation, and how this contributes to the efficiency of inferring common and unique components from data?


**Limitations:**

The authors address the limitation adequately.

---

> ### Author Rebuttal · Authors · 2023-08-09
>
> We thank the reviewer for their comments and address their questions below.
>
> > The authors experimentally demonstrating that adding unsupervised viewpoints improves disentanglement; which is likely due to limitations in single-sample analysis; active interaction, not passive observation, fosters better learning of environment representations. but the observation has not been sufficiently explained or explored in the current study.
>
> We agree with the reviewer’s observation and intuition; and we will happily elaborate on this in the text. Indeed, as we briefly mentioned in the text, a source of inspiration for our work was an old experiment where neuroscientists showed that the ability for animals to act within an environment, rather than passively observe it strongly impacts their representations of it and ability to identify relevant environmental variables (Held & Hein, 1963 ). One explanation for these findings is that the correlations between the actions and the environmental stimuli are meaningful and important for the animal to leverage.  Here, in a similar spirit, we indeed found that the addition of an additional viewpoint can provide a guiding signal for finding the common information between the views, which can identify and reveal interesting structure in the data.
>
> > Could you describe how the multi-view VAE model's unique latent variables capture the individual factors that cause variation, and how this contributes to the efficiency of inferring common and unique components from data?
>
> In our optimization, adding the unique latent variables (with the constraint that $\beta_u > \beta_c$) was important so that unique information would not be encoded in the common latent factors. While this is penalized by $\lambda_c$, as well as by randomly sampling $z$ from $q_1(z_c|x_1)$ and $q_2(z_c|x_2)$ with $p = 0.5$, without the addition of the unique components, the unique information could leak into the common latents if the unique information helped for reconstructing the output more than the cost of not satisfying the (approximate) equality constraint. By adding in the unique latent variables (with $\beta_u > \beta_c$) the common and unique information will be partitioned, as we show in Thm 3.1 and our experiments.
>
> ---
> Richard Held and Alan Hein. Movement-produced stimulation in the development of visually guided behavior. Journal of comparative and physiological psychology, 56(5):872, 1963.

---

> > ### Comment · Reviewer_cgd1 · 2023-08-17
> >
> > Thanks for your responses and explanations.  I keep my score as it is.

---

### Official Review · Reviewer_iAUJ · 2023-07-07

**Soundness:** 3 good
**Presentation:** 2 fair
**Contribution:** 2 fair
**Rating:** 5
**Confidence:** 4

**Summary:**

The basis of the proposed work is a practical means to decompose the information contained in two random variables into common and unique components.  With a couple tweaks to a vanilla VAE setup, data that has been paired such that certain factors of variation have a constant value within the pair can be used to train latent spaces that separate the components of the information.  The common component of the information -- the Gacs-Korner common information -- is the primary goal here, and it’s generally difficult to acquire because it requires finding a random variable that is simultaneously a deterministic function of both input variables.  Except for highly constrained joint distributions, such a variable generally does not exist (beyond the vacuous solution).  The current work proposes a relaxation where the common component can be a stochastic function of the inputs, and this facilitates optimization as well as makes the method applicable to cases without cleanly shared information (at least in theory; it’s not shown here).

As a route to Gacs-Korner common information, the paper offers a principled and pragmatic methodology that (as far as I can tell) offers something original and of value.  The example scenarios and applied metrics, however, paint the work more as a route to partially disentangled latent spaces given grouped data, and in that regard it’s lacking novelty and missing relevant comparisons.  The following papers leverage the same weak supervision (paired/grouped data with a subset of factors held constant) to learn latent spaces that separate common from unique information:

- “Multi-Level Variational Autoencoder: Learning Disentangled Representations from Grouped Observations”, Bouchacourt 2018
- “Disentangling Factors of Variation with Cycle-Consistent Variational Auto-Encoders”, Jha 2018
- “Unsupervised Robust Disentangling of Latent Characteristics for Image Synthesis”, Esser 2019
- “NestedVAE: Isolating Common Factors via Weak Supervision” Vowels 2020

And one that does so without a generative model:
- “Learning Disentangled Representations via Mutual Information Estimation”, Sanchez 2020

In summary, the method is an interesting contribution but the experimental results are not very effective.


**Strengths:**

The principled route to extracting the common information in a pragmatic methodology is great.  Though it is not demonstrated in this work, the authors motivate the work in terms of multi-modality data, which could be a rich area of application of the method.

**Weaknesses:**

See the summary for the primary weakness.  I can imagine a couple routes to make the experimental results more strongly support the method.  First, the current experiments could include direct comparisons to some of the attached methods with a demonstration of what GKVAE brings.  However, if GKVAE is not more effective in the scenarios currently in the manuscript, qualitatively different experiments could help show its merit.  Paired data streams from different modalities could highlight the strengths of GKVAE, or perhaps an example where the content of the common information allows for insight about the relationship between the variables (ie, through inspection of the learned latent variable).  Or some example where GK information is desirable.

On a much more minor scale: I doubt the majority of readers will have a better intuition for “usable information” than for standard mutual information; it seems a weakness to me to rely on that rather than any of a number of mutual information estimators or bounds that should work fine on the low-entropy examples of this work.


**Questions:**

- As common information is a lower bound to mutual information (L73), how can a lower bound to MI be used as a lower bound for common information (L194)?
- Was $\lambda_c$ shown to be necessary given the random swapping of the common representation (L225)?

**Limitations:**

Yes

---

> ### Author Rebuttal · Authors · 2023-08-09
>
> We thank the reviewer for their comments and address their concerns below.
>
> To understand the **difference between GK information and mutual information and why GK information is desirable**, consider the following settings.
>
> **Case 1:**
> Let $X_1 = C + N_1$ and $X_2 = C + N_2$, where $C$ and $N_i$ are independent variables, but the noise $N_1$ and $N_2$ are correlated.
> Suppose $C$ is discrete (e.g. a 0 or 1 feature), and have $N_i$ be a relatively small amount of gaussian noise. Now $C$ can still be recovered deterministically from $C + N_i$ (just threshold) but the mutual information between $C + N_1$ and $C + N_2$ is $1 + 0.5 \log (\frac{\det(\Sigma_1) \det(\Sigma_2) } {\det(\Sigma)} )$ (the first bit comes from the MI of $C$, the log comes from the MI between gaussian variables). Then the MI can be made arbitrarily large by increasing the correlation between $N_1$ and $N_2$ (making $\det(\Sigma) \to 0$).
>
> In contrast, the GK common information only identifies the common component C. In a sense, the GK is able to recover the underlying "discrete" or "symbolic" information that is common between two different continuous sources. This is a valuable property to have in settings like neuroscience where one wants to measure what is actually encoded by both representations and not just how much they are correlated (which could be due to any amount of spurious factors). Additionally, with our relaxation of the problem we can measure how much is almost encoded in both, which makes the method more applicable to real-world cases where it is unlikely the same exact thing is encoded in both.
>
> **Case 2:**
> Let  $X_1 = f_1(C; U_1)$ and $X_2 = f_2(C; U_2)$ where $C$ and $U_i$ are independent but $U_1$ and $U_2$ are correlated. Suppose $f_i$ is an invertible data generating function mapping latent factors to observations (such as pixels). By maximizing mutual information between views, one is implicitly caring about those correlations (based on a similar computation to above in Case 1), whereas in the GK sense one cares about only the fully shared components between views. These different perspectives can offer different utility depending on the use case.
>
> We will clarify this distinction in the main text, which we hope is helpful for readers. In addition to the benefits of interpreting a well-defined common core between variables, the GK common information can be helpful for compression schemes, as we mention in the discussion.
>
> We appreciate the reviewer noting that our contribution is interesting in bridging the efforts in information theory to define a well-defined notion of common information and recent progress in machine learning to exploit such knowledge coming from multiple data streams. We agree that there is a growing literature on multi-view representation learning that aims to partition the information between views (which we discuss in part in our paper) and we will update our citations with the references you mention which share a similar goal -- thank you for suggesting them! The existing methods appear to focus on qualitatively partitioning the information through the use of different objectives, however, the setups appear to be difficult to prove theorectical guarantees (as they lack a well-defined definition) or enable quantification of the information contained in the representation. In contrast, our setup allows us to formalize the decomposition of information using a well-defined information theoretic notion of common/unique information and we show both formally and empirically that we can optimize the objective with a simple modification to a traditional VAE setup, and also enables quantification of such information. We compare our method against contrastive training, finding that contrastive approaches only identify the common information (Fig. 3).
>
> We also agree with the reviewer that the ultimate goal is to leverage our framework for application on real-world high dimensional multi-modal data, or for interpreting high dimensional scientific data (such as neural recordings from different brain areas) and are excited about the insights this can yield. We believe showing that a well-defined partitioning of information exists and can be approximated from data using our framework is an important step to this goal. For example, accurately modeling for example high-resolution RGB + Depth data requires task specific architectures, and we leave it to future work, and we believe our work is an important step towards this ultimate goal.
>
> Q1: We are looking at mutual information contained in the representation that satisfies the constraint that it is the common information (approximately). We then look at the usable/mutual information that the representation contains about the ground-truth latent variable.
>
> Q2: In terms of the necessity of $\lambda_c$, the random sampling of the latent may not be enough to enforce that only common information is encoded in $z_c$, as in principle $z_c$ could encode unique information about a view that is then discarded by the view-specific decoder. The constraint on $\lambda_c$ guarantees that that will not be the case.  In Fig. 1 in the rebuttal material, we run an ablation experiment setting $\lambda_c=0$. We observe that the implicit bias due to the random swapping suggested by the reviewer is enough to have a qualitative separation between common and unique information in the experiment (Fig. 1, left). However, in Table 1 of the rebuttal we observe that in this same setting the divergence between the distribution $q_1(z_c|x_1)$ and $q_2(z_c|x_2)$ was exceedingly large (7.5 * 10^5) which would lead to severe mis-estimation of the GK common information. Increasing the value of $\lambda_c$ solves the problem (Table 1 of rebuttal material). Higher-values of $\lambda_c$ will increasingly satisfy the constraint of the GK information, but may destabilize training. Empirically, we found that $\lambda_c=0.1$ provided a good trade-off in our experiments.

---

> > ### Comment · Reviewer_iAUJ · 2023-08-18
> >
> > Thank you for the effort put into the rebuttal and the clarity on the role of $\lambda_c$ provided by the additional experiments.
> >
> > After consideration of the rebuttal and the other reviews, I am still of the opinion that the premise of this work is interesting and valuable but the experimental support is ineffective. I strongly recommend more relevant baselines on the current experiments and/or use cases where the distinction between common and shared/mutual information is more meaningful -- even a simple synthetic one as suggested in the above response would be helpful.  GK common information is far enough off the beaten path that the onus lies heavily on the authors to demonstrate why the goal is worthwhile, especially in the midst of a variety of other methods that accomplish qualitatively similar separation of information.

---

> > > ### Author Response · Authors · 2023-08-22
> > >
> > > Thank you for your comment and feedback! In line with the use cases we described in the above response, we plan to include a supporting synthetic experiment to better highlight the use cases and the difference between Gacs-Korner common information and mutual in the camera-ready version.

---

> ### Comment · Area_Chair_XDqo · 2023-08-18
> **Acknowledging the rebuttal**
>
> Dear reviewer,
>
> Thank you for your time and effort.
>
> The authors have tried to address you comments in their rebuttal.
> What do you think about their response?
>
> Could you please acknowledge the rebuttal as well as the other reviews.
>
> Thanks,
>
> The AC

---

### Official Review · Reviewer_hfAm · 2023-07-07

**Soundness:** 2 fair
**Presentation:** 3 good
**Contribution:** 3 good
**Rating:** 5
**Confidence:** 3

**Summary:**

The paper proposes the Gracs-Korner Common information to measure common information between two random variables and propose a variational relaxation to compute the practical loss. The paper then derives an objective that eases disentanglement requirement.

**Strengths:**

1. The paper proposes the Gracs-Korner Common information to measure common information between two random variables and propose a variational relaxation to compute the practical loss. To this point, the paper is novel to me. Empirical evidence demonstrates the advantages of the proposed method.

**Weaknesses:**

1. The proposed method is only compared with basic VAE where as the advantage of the proposed method is not significant.

2. The proposed method seems to be loosely connected with Gacs-Korner Common Information. The derivation is a minor variation of a vanilla VAE. In this regard, it seems the impact of the work is limited, where the modification introduces additional encoder architecture but only brings limited empirical advantage. The final loss further decompose the distance between prior and posterior inference into “common components-prior distances” and the “unique components-prior distances". The driven force that separates the DNN to learn the decomposition remains unclear.

**Questions:**

Please see weakness above.

**Limitations:**

Yes.

---

> ### Author Rebuttal · Authors · 2023-08-09
>
> We thank the reviewer for their comments and address their concerns below.
>
> > The proposed method is only compared with basic VAE where as the advantage of the proposed method is not significant.
>
> The purpose of our study was to show formally and empirically that we can partition the latent representation of multi-view data into a common and unique component, and also provide a tractable approximation for the Gacs-Korner common information between high dimensional random variables. In comparison against a (Beta)-VAE, in addition to our empirically observed higher disentanglement scores, the (Beta)-VAE cannot provide insight into which factors are shared and which are unique, whereas our optimization directly provides this separation. Additionally, even theoretically it has been shown that VAE cannot disentangle components without additional signals (like, in our case, the common information between different sensors) (Locatello et al., 2019).
>
> > The proposed method seems to be loosely connected with Gacs-Korner Common Information. The derivation is a minor variation of a vanilla VAE. In this regard, it seems the impact of the work is limited, where the modification introduces additional encoder architecture but only brings limited empirical advantage. The final loss further decompose the distance between prior and posterior inference into “common components-prior distances” and the “unique components-prior distances". The driven force that separates the DNN to learn the decomposition remains unclear.
>
> In Theorem 3.1, we prove that our notion of common information is a direct generalization of the original Gacs-Korner common information. Moreover, we show that our variational relaxation of the problem provides a tractable way to compute the common information between high-dimensional random variables, which is a major obstacle with the original proposal of GK-Common Information. We show that our approach recovers the correct value of the GK common information in all benchmarks. Moreover, we introduce for the first time a series of benchmarks with high-dimensional random variables (which previous algorithms could not address) and we show that even here we can obtain the correct theoretical value of the common information.
>
> Most results in structured representation learning with VAE involve loss functions modified with additional constraints. We believe that the fact that the changes need to decompose the representation in common and unique factors are relatively simple to implement, theoretically grounded, and widely applicable to any existing VAE architecture are an advantage of our method, not a downside.
>
> The key parameters enabling this optimization are the relative values of $\beta_u$ and $\beta_c$, with $\beta_u > \beta_c$, which encourages common information to be encoded in the common latents by paying a smaller cost. $\lambda_c$ encourages the distributions of the encoders to be similar so that the factor encodes common information (we additionally bias the information to be common through random sampling of the common latents from either encoder during training).
>
> ---
>
> Locatello et al. Challenging common assumptions in the unsupervised learning of disentangled representations. ICML, 2019

---

> > ### Comment · Reviewer_hfAm · 2023-08-20
> > **thanks for your reply**
> >
> > Thanks for the reply! I will maintain my score.
> >
> > Best

---

> ### Comment · Area_Chair_XDqo · 2023-08-18
> **Acknowledging the rebuttal**
>
> Dear reviewer,
> Thank you for your time and effort.
>
> The authors have tried to address you comments in their rebuttal.
> What do you think about their response?
>
> Could you please acknowledge the rebuttal as well as the other reviews.
>
> Best,
>
> The AC

---

### Official Review · Reviewer_uZkW · 2023-07-15

**Soundness:** 3 good
**Presentation:** 2 fair
**Contribution:** 3 good
**Rating:** 5
**Confidence:** 4

**Summary:**

This paper proposes a new way to partition common and unique information in multi-view data, by leveraging and optimizing the objective of common information defined by Gacs and Korner. Authors extended the deterministic setup to stochastic scenario and used a variational autoencoder to realize the objective. Authors also carefully designed experiments in both static and time series data to validate the effectiveness of their architecture.


**Strengths:**

1. The introduction of the Gacs-Korner common information to partition common and unique information in realistic multi-view data is novel.
2. A straightforward application of Gacs-Korner common information is hard. The stochastic relaxation made by authors makes sense to me.
It is also good that authors constructed different datasets to validate the effectiveness of their method.

**Weaknesses:**

 I found the biggest issue of this manuscript is that some points are very unclear or need more explanations. Below I listed a few of them:
1) Can you elaborate more on the difference between common information and mutual information.
I can understand that mutual information has no clear interpretation in terms of information decomposition.
It would be much better to describe more differences.
For example, in Eq.~(15), what would be the mutual information between $x_1$ and $x_2$; and why mutual information should be much larger than common information $z_c$?
Are there more illustrative examples?
2) In terms of implementation, I appreciate that authors relax the constraint of $Z=f(X_1)=g(X_2)$ with a conditional divergence minimization term $D(p(z|x_1);p(z|x_2))$.
How to implement this term in your VAE objective?
3) Also in implementation, it requires that $\beta_u > \beta_c$. How to balance the trade-off between $\lambda_c$ and $\beta$?
In Eq.~(13), there should be two $\beta_u$ corresponding to two views, i.e., $\beta_{u1} KL(q_{\phi_{u1}}||...)+ \beta_{u2} KL(q_{\phi_{u2}}||...)$?
4) Proposition 3.1 requires an invertible mapping from $z$ to $x$. How is this condition reflected in your implementation?

**Questions:**

Please refer to weaknesses above.

**Limitations:**

Authors did not discuss potential limitations and negative societal impacts.

---

> ### Author Rebuttal · Authors · 2023-08-09
>
> We thank the reviewer for their feedback and address their concerns below.
>
> Q1: We elaborate on the difference between common information and mutual information in our response to [Reviewer iAUJ](https://openreview.net/forum?id=e4XidX6AHd&noteId=HqfjeyTDaR) (see Case 1 and Case 2 in particular).
>
> Q2: This was achieved through a combination of using $\lambda_c$ and also sampling independently from both encoders with $p=0.5$ (Line 226). As per the request of reviewer iAUJ, we ran the ablation with $\lambda_c = 0$ and found that the implicit bias due to the random swapping suggested by the reviewer is enough to have a weak qualitative separation between common and unique information (Fig. 1 of rebuttal pdf). However, in Table 1 of the rebuttal we observe that in this same setting the divergence between the distribution and was exceedingly large (7.5 * 10^5) which would lead to severe mis-estimation of the GK information in the common component. Increasing the value of solves the problem (Table 1 of rebuttal material). Higher-values of will increasingly converge to the exact GK information, but may destabilize training. Empirically, we found that $\lambda_c = 0.1$ provided a good trade-off.
>
> We parametrized the encoders $p(z|x_1)$ and $p(z|x_2)$ using neural networks that outputted the mean and (diagonal) covariance of a Gaussian distributuion. In this case, the KL divergence (which we denote by D) has a close form, which we minimize during training.
>
> Q3: Related to the above, the purpose of $\lambda_c$ was to ensure that the information was common. We found that we could set $\lambda_c$ to be a large value provided we started it off at a small value at the beginning of training (Fig. 2 of rebuttal).
> In this way, the effect of the parameter $\lambda_c$ appears to depend more strongly on it's value during training rather than the values of $\beta_u$ and $\beta_c$.
>
> Indeed there should be two $\beta_u$ in Eq.~3 corresponding to the two views (in our experiments they were equal). We will clarify the text.
>
> Q4: This assumption refers to condition on the data generating model so that the latents can be recovered: it’s saying that we need to be able to recover the latents from the data in order for our optimization to recover the ground-truth latents. In the datasets we examine, this appears to hold, as indeed we can decode the common and unique latents from the common and unique factors. Note that even if this assumption does not hold in practice, our method will still recover the common information and unique information present in the observed data.
>
> We discussed limitations in Appendix D, and we do not foresee any negative societal impacts.

---

> > ### Comment · Reviewer_uZkW · 2023-08-17
> >
> > Thanks for your rebuttal. I do not have more concerns. It is good that authors use to two illustrative examples to demonstrate the merits of CI over MI. I suggest including this discussion also in the main text. My score remains the same, since I agree with Reviewer iAUJ that there are growing number of literatures on multi-view (disentangled) representation learning, which may also define their own ways to separate unique and common information. I understand that those literatures may suffer from a rigorous definition on what is common and what is unique. However, a comparison to state-of-the-art can enhance the quality of this paper.

---

### Author Rebuttal · Authors · 2023-08-09

We thank all the reviewers for their thoughtful comments and suggestions, which we believe will lead to a clearer and improved manuscript.

We have responded to each of the reviewer's questions individually. Please see the attached pdf for figures and a table that we refer to in our responses to specific reviewers.

---

### Decision · Program_Chairs · 2023-09-21

**Decision:**

Accept (poster)

**Comment:**

This paper proposes a new way to partition common and unique information in multi-view data, by leveraging and optimizing the objective of common information defined by Gacs and Korner. The paper extends the deterministic setup to stochastic scenario and uses a variational auto-encoder to achieve this objective. Carefully designed experiments in both static and time series data validate the effectiveness of the method.


I agree with the reviewers that the contributions are sound and novel and would be of interest to the community.